# Contextual Bandits with Knapsacks
# for a Conversion Model

**Zhen Li**
BNP Paribas, 16 boulevard des Italiens, 75009 Paris, France
zhen.li@bnpparibas.com

**Gilles Stoltz**
Université Paris-Saclay, CNRS, Laboratoire de mathématiques d'Orsay, 91405, Orsay, France
gilles.stoltz@universite-paris-saclay.fr
HEC Paris, 1 rue de la Libération, 78350 Jouy-en-Josas, France
stoltz@hec.fr

## Abstract

We consider contextual bandits with knapsacks, with an underlying structure between rewards generated and cost vectors suffered. We do so motivated by sales with commercial discounts. At each round, given the stochastic i.i.d. context $\boldsymbol{x}_t$ and the arm picked $a_t$ (corresponding, e.g., to a discount level), a customer conversion may be obtained, in which case a reward $r(a, \boldsymbol{x}_t)$ is gained and vector costs $\boldsymbol{c}(a_t, \boldsymbol{x}_t)$ are suffered (corresponding, e.g., to losses of earnings). Otherwise, in the absence of a conversion, the reward and costs are null. The reward and costs achieved are thus coupled through the binary variable measuring conversion or the absence thereof. This underlying structure between rewards and costs is different from the linear structures considered by Agrawal and Devanur [2016] (but we show that the techniques introduced in the present article may also be applied to the case of these linear structures). The adaptive policies exhibited solve at each round a linear program based on upper-confidence estimates of the probabilities of conversion given $a$ and $\boldsymbol{x}$. This kind of policy is most natural and achieves a regret bound of the typical order $(\text{OPT}/B)\sqrt{T}$, where $B$ is the total budget allowed, OPT is the optimal expected reward achievable by a static policy, and $T$ is the number of rounds.

## 1 Introduction and Literature Review

We consider the framework of stochastic multi-armed bandits, which has been extensively studied since the early works by Thompson [1933] and Robbins [1952]. Two recent (and complementary) surveys summarizing the latest research in the field were written by Lattimore and Szepesvári [2020] and Slivkins [2019]. On the one hand, we are particularly interested in the setting of *contextual* stochastic multi-armed bandits, preferably with some structural assumptions on the dependency between rewards and contexts: linear models (again, a rich literature, see, among many others, Chu et al. [2011] and Abbasi-Yadkori et al. [2011], whose work marked a turning point), and, for $[0, 1]$-valued rewards, logistic models (Filippi et al. [2010] and Faury et al. [2020]). On the other hand, we are also particularly interested in stochastic multi-armed bandits *with knapsacks*, i.e., with cumulative vector-cost constraints to be abided by on top of maximizing the accumulated rewards. The setting was introduced by Badanidiyuru et al. [2013, 2018] and a comprehensive summary of the results achieved since then may be found in Slivkins [2019, Chapter 10]. The intersection of these two frameworks of interest is called *contextual bandits with knapsacks* [CBwK] and is the focus of the present article.

36th Conference on Neural Information Processing Systems (NeurIPS 2022).

**Literature review on CBwK.** The first approach to CBwK, by Badanidiyuru et al. [2014] and Agrawal et al. [2016], assumes a joint stochastic generation of triplets of contexts-rewards-costs, with no specific underlying structure, and makes the problem tractable by using as a benchmark a finite set of static policies. As noted by Agrawal and Devanur [2016], picking this finite set may be uneasy, which is why they introduce instead a structural assumption of linear modeling: the (unknown) expected rewards and cost vectors depend linearly on the contexts.

We consider a different modeling assumption, motivated by sales with commercial discounts (see Appendix A): general (known) reward and cost functions are considered but they are coupled via a 0/1–valued factor, called a (customer) conversion, obtained as the realization of a Bernoulli variable with parameter $P(a, \boldsymbol{x})$ depending on the context $\boldsymbol{x}$ observed (customer's characteristics) and the action $a$ taken (discount level offered). The probabilities $P(a, \boldsymbol{x})$ are themselves modeled by a logistic regression, whose parameters may be learned through an adaptation of the techniques by Filippi et al. [2010] and Faury et al. [2020]. We do so in the first phase of the adaptive policy introduced in this article. More details on the comparison of the new setting considered to known settings of CBwK may be found in Section 2.2.

**Primal-dual approach.** The second phase of the adaptive policy exhibited uses the primal-dual approach to a convex optimization problem—actually, a simple optimization problem given by a linear program. This approach was already used in various ways for bandits with knapsacks, including CBwK, to define policies based on the dual problem: this is explicit in the LagrangeBwK policy of Immorlica et al. [2019] and is implicit in the reward-minus-weighted-cost approach of Agrawal and Devanur [2016] and Agrawal et al. [2016], as we underline in the proof sketch of Section 4 as well as in the discussion of Section 6. However, we only use the primal-dual approach in the analysis and state our adaptive policy directly in terms of the primal problem, where we substituted upper-confidence estimates of the probabilities $P(a, \boldsymbol{x})$. We therefore end up with a most natural adaptive policy, which mimics the optimal static policy used as a benchmark. This direct primal statement of the policy actually also works for the setting of linear CBwK studied by Agrawal and Devanur [2016], as we show in Section 6. Policies based on such direct primal statements were already considered for bandits with knapsacks (see Li et al. [2021] and references therein) but do not seem easily extendable to CBwK.

**Outline and main contributions.** The first contribution of this article is a new structured setting of CBwK, based on a coupling between general rewards and cost vectors through conversions modeled based on a logistic regression; we present and discuss it in Section 2.1 (and explain its origins in Appendix A of the supplementary material). The adaptive policy introduced is described in Section 3. Its first phase consists of learning the parameter of logistic regression and is adapted from Faury et al. [2020]. Its second phase—and this is the second contribution of this article—directly solves a primal problem with optimistic conversion probabilities. The analysis, which we believe is concise, elegant, and natural, is provided in Sections 4 (when the context distribution $\nu$ is known) and 5 (when $\nu$ is unknown). As mentioned above, Section 6 draws the consequences of our second contribution for linear CBwK.

**Notation.** Throughout the article, vectors are denoted with bold symbols. In particular, $\mathbf{0}$ and $\mathbf{1}$ denote the vectors with all components equal to $0$ and $1$, respectively. With no additional subscript, $\|\mathbf{v}\|$ denotes the Euclidean norm of a vector $\mathbf{v}$, while a subscript given by a non-negative symmetric matrix $M$ refers to $\|\mathbf{v}\|_M = \sqrt{\mathbf{v}^\mathsf{T} M \mathbf{v}}$.

## 2 Learning Protocol and Motivation

We describe the learning protocol and objectives considered (Section 2.1) and explain why it is not covered by earlier works (Section 2.2). We also detail (Appendix A in the supplementary material) how this learning protocol was defined based on an industrial motivation in the banking sector: market share expansion for loans by granting discounts, under commercial budget constraints.

### 2.1 Learning Protocol and Modeling Assumptions

We consider a finite action set $\mathcal{A}$, including a special action $a_{\text{null}}$ called no-op, and a finite context set $\mathcal{X} \subseteq \mathbb{R}^n$. (We discuss and mitigate finiteness of $\mathcal{X}$ in Section 2.2.) A scalar reward function $r : \mathcal{A} \times \mathcal{X} \to [0, 1]$ and a vector-valued cost function $\boldsymbol{c} : \mathcal{A} \times \mathcal{X} \to [0, 1]^d$ evaluate the performance

of actions given the contexts. There are several sources of costs to control: each corresponds to a component of $\boldsymbol{c}$. We assume that these function are known, and (with no loss of generality) that their ranges are $[0, 1]$ and $[0, 1]^d$. The no-op action induces null reward and costs: $r(a_{\text{null}}, \boldsymbol{x}) = 0$ and $\boldsymbol{c}(a_{\text{null}}, \boldsymbol{x}) = \boldsymbol{0}$ for all $\boldsymbol{x} \in \mathcal{X}$.

Contexts—which correspond, for instance, to customers' characteristics, see Appendix A—are drawn sequentially according to some distribution $\nu$, which may be known or unknown (we will deal with both cases). At each round $t \geqslant 1$, upon observing the context $\boldsymbol{x}_t \in \mathcal{X}$ drawn, the learner picks an action $a_t \in \mathcal{A}$, which corresponds, for instance, to an offer made to the customer $t$. If the latter accepts the offer, an event which we denote $y_t = 1$, then the learner obtains a reward $r(a_t, \boldsymbol{x}_t)$ and suffers some costs $\boldsymbol{c}(a_t, \boldsymbol{x}_t)$. When the customer declines the offer, we set $y_t = 0$, and null reward and costs are obtained. Thus, in both cases, the reward and costs may be written as $r(a_t, \boldsymbol{x}_t) y_t$ and $\boldsymbol{c}(a_t, \boldsymbol{x}_t) y_t$. We call $y_t$ the conversion and now explain how it is modeled.

**Modeling conversions.** We model each conversion $y_t$ as an independent Bernoulli random drawn, with parameter $P(a_t, \boldsymbol{x}_t)$ depending on the context $\boldsymbol{x}_t$ and action $a_t \neq a_{\text{null}}$. We further assume that these probabilities may be written as a logistic regression model, i.e., there exists a known transfer function $\boldsymbol{\varphi} : \mathcal{A} \setminus \{a_{\text{null}}\} \times \mathcal{X} \to \mathbb{R}^m$ and some unknown parameter $\boldsymbol{\theta}_\star \in \mathbb{R}^m$ such that

$$\forall \boldsymbol{x} \in \mathcal{X}, \ \forall a \in \mathcal{A} \setminus \{a_{\text{null}}\}, \qquad P(a, \boldsymbol{x}) = \eta\big(\boldsymbol{\varphi}(a, \boldsymbol{x})^{\mathrm{T}} \boldsymbol{\theta}_\star\big), \qquad \text{where} \quad \eta(x) = 1/(1 + \mathrm{e}^{-x}) \, . \ (1)$$

We assume that $\boldsymbol{\varphi}$ is normalized in a way that its Euclidean norm satisfies $\|\boldsymbol{\varphi}\| \leqslant 1$ and that a bounded convex set $\Theta$ containing $\boldsymbol{\theta}_\star$ is known. Such a modeling is natural and opens the toolbox of logistic bandits; see Faury et al. [2020] and references cited therein. We however note (and discuss this fact in Appendix C) that the logistic regression model above is slightly different from the one by Faury et al. [2020].

The concept of a conversion $y$ for a round when the no-op action $a_{\text{null}}$ is played is void, and thus, we leave the probabilities $P(a_{\text{null}}, \boldsymbol{x})$ undefined, though by an abuse of notation, these quantities might appear but always multiplied by a 0, given, e.g., by indicator functions like $\mathbb{1}_{\{a \neq a_{\text{null}}\}}$, null rewards $r(a_{\text{null}}, \boldsymbol{x})$, or null costs $\boldsymbol{c}(a_{\text{null}}, \boldsymbol{x})$.

**Policies: static vs. adaptive.** The learner is given a number of rounds $T$ and a maximal budget $B$ (the same for all cost components, with no loss of generality: up to some normalization). A static policy is a function $\pi : \mathcal{X} \to \mathcal{P}(\mathcal{A})$, where $\mathcal{P}(\mathcal{A})$ is the set of probability distributions over $\mathcal{A}$. As is traditional in the literature of CBwK (we recall below why this is the case), we take as benchmark the static policy $\pi^\star$ with largest expected cumulative rewards under the condition that its cumulative costs abide by the budget constraints in expectation. More formally, $\pi^\star$ achieves the maximum defining

$$\text{OPT}(\nu, P, B) = \max_{\pi : \mathcal{X} \to \mathcal{P}(\mathcal{A})} T \, \mathbb{E}_{\boldsymbol{X} \sim \nu} \left[ \sum_{a \in \mathcal{A}} r(a, \boldsymbol{X}) \, P(a, \boldsymbol{X}) \, \pi_a(\boldsymbol{X}) \right]$$

$$\text{under} \qquad T \, \mathbb{E}_{\boldsymbol{X} \sim \nu} \left[ \sum_{a \in \mathcal{A}} \boldsymbol{c}(a, \boldsymbol{X}) \, P(a, \boldsymbol{X}) \, \pi_a(\boldsymbol{X}) \right] \leqslant B \boldsymbol{1} \, , \tag{2}$$

where $\mathbb{E}_{\boldsymbol{X} \sim \nu}$ denotes an expectation solely over random contexts $\boldsymbol{X}$ following distribution $\nu$, where $\pi_a(\boldsymbol{X})$ denotes the probability mass put by $\pi(\boldsymbol{X})$ on $a \in \mathcal{A}$, and where $\leqslant$ is understood component-wise. Of course, the sums in the two expectations above are taken indifferently over $\mathcal{A}$ or $\mathcal{A} \setminus \{a_{\text{null}}\}$.

The learner uses an adaptive policy, i.e., a sequence of measurable functions $\boldsymbol{p}_t : \mathcal{H}^{t-1} \times \mathcal{X} \to \mathcal{P}(\mathcal{A})$ indexed by $t \geqslant 1$, where $\mathcal{H} = \mathcal{X} \times \mathcal{A} \times \{0, 1\}$. Indeed, the history available to the learner at the beginning of the round $t \geqslant 2$ is summarized by $h_{t-1} = (\boldsymbol{x}_s, a_s, y_s)_{s \leqslant t-1}$, and we define $h_0$ as the empty vector. Such a policy draws the action $a_t$ for round $t \geqslant 1$ independently at random according to $\boldsymbol{p}_t(h_{t-1}, \boldsymbol{x}_t)$. We impose hard budget constraints on adaptive policies: they must satisfy

$$\sum_{t \leqslant T} \boldsymbol{c}(a_t, \boldsymbol{x}_t) y_t \leqslant B \boldsymbol{1} \quad \text{a.s.}$$

Such adaptive policies are called feasible in the literature. To abide by these hard constraints, we may restrict our attention to adaptive policies that pick Dirac masses on $a_{\text{null}}$ whenever one component of the cumulative costs is larger than $B - 1$. At the same time, an adaptive policy should maximize the cumulative rewards obtained or, equivalently, minimize its regret:

$$R_T = \text{OPT}(\nu, P, B) - \sum_{t \leqslant T} r(a_t, \boldsymbol{x}_t) y_t \, .$$

> **BOX A: CONTEXTUAL BANDITS WITH KNAPSACKS [CBwK] FOR A CONVERSION MODEL**
>
> **Known parameters:** finite action set $\mathcal{A}$ including a no-op action $a_{\text{null}}$; finite context set $\mathcal{X} \subseteq \mathbb{R}^n$; scalar reward function $r : \mathcal{A} \times \mathcal{X} \to [0,1]$; vector-valued cost function $\boldsymbol{c} : \mathcal{A} \times \mathcal{X} \to [0,1]^d$; number $T$ of rounds; total budget constraint $B > 0$.
>
> **Possibly unknown parameters:** context distribution $\nu$ on $\mathcal{X}$; probability of conversion given action and context $P : \mathcal{A} \setminus \{a_{\text{null}}\} \times \mathcal{X} \to [0,1]$, modeled as $P(a, \boldsymbol{x}) = \eta\big(\boldsymbol{\varphi}(a, \boldsymbol{x})^{\mathsf{T}} \boldsymbol{\theta}_\star\big)$ for some known transfer function $\boldsymbol{\varphi} : \mathcal{A} \setminus \{a_{\text{null}}\} \times \mathcal{X} \to \mathbb{R}^m$, with $\|\boldsymbol{\varphi}\| \leqslant 1$, and some unknown parameter $\boldsymbol{\theta}_\star \in \mathbb{R}^m$, lying in a known bounded convex set $\Theta$.
>
> **For rounds** $t = 1, 2, 3, \ldots, T$:
> 1. Context $\boldsymbol{x}_t \sim \nu$ is drawn independently of the past;
> 2. Learner observes $\boldsymbol{x}_t$ and picks an action $a_t \in \mathcal{A}$;
> 3. Conversion $y_t \in \{0, 1\}$ is drawn according to $\text{Ber}\big(P(a_t, \boldsymbol{x}_t)\big)$;
> 4. Learner observes $y_t$, gets reward $r(a_t, \boldsymbol{x}_t)\, y_t$, and suffers costs $\boldsymbol{c}(a_t, \boldsymbol{x}_t)\, y_t$.
>
> **Goals:** Maximize $\displaystyle\sum_{t \leqslant T} r(a_t, \boldsymbol{x}_t)\, y_t$ while controlling $\displaystyle\sum_{t \leqslant T} \boldsymbol{c}(a_t, \boldsymbol{x}_t)\, y_t \leqslant B\mathbf{1}$

It may be proved (along the same lines as Agrawal and Devanur [2016, Appendix B] do for a different model) that the optimal static policy $\pi^\star$ obtains, on average and in expectation, a cumulative reward at least as good as the best feasible adaptive policy.

**Summary.** A summary of the learning protocol and of the goals is provided in Box A. We note here that rewards gained and vector costs suffered at round $t$ in the case $y_t = 1$ of a conversion could be stochastic with expectations $r(a_t, \boldsymbol{x}_t)$ and $\boldsymbol{c}(a_t, \boldsymbol{x}_t)$: our analysis and the regret bounds would be unchanged, as long as the expectation functions $r$ and $\boldsymbol{c}$ are known.

## 2.2 Discussion and Comparison to Existing Learning Protocols

The setting described above may be reduced to the general setting of CBwK, as introduced by Badanidiyuru et al. [2014] and Agrawal et al. [2016]. Indeed, introduce independent Bernoulli variables $y_{t,a}$ with parameters $P(a, \boldsymbol{x}_t)$, for all $a \in \mathcal{A} \setminus \{a_{\text{null}}\}$, and set $y_{t,a_{\text{null}}} = 0$. The vectors

$$\Big(\boldsymbol{x}_t, \big(r_t(a)\big)_{a \in \mathcal{A}}, \big(\boldsymbol{c}_t(a)\big)_{a \in \mathcal{A}}\Big), \qquad \text{where} \quad r_t(a) = r(a, \boldsymbol{x}_t)\, y_{t,a} \quad \text{and} \quad \boldsymbol{c}_t(a) = \boldsymbol{c}(a, \boldsymbol{x}_t)\, y_{t,a}$$

are i.i.d., and upon picking action $a_t \in \mathcal{A}$, the obtained and observed rewards and cost vectors equal $r_t(a_t)$ and $\boldsymbol{c}_t(a_t)$. When $\mathcal{X}$ is discrete, we may consider the set $\Pi$ of base policies that map $\mathcal{X}$ to $\{\delta_a : a \in \mathcal{A}\}$, the set of Dirac masses at some $a \in \mathcal{A}$. The convex hull of $\Pi$ is the set of all static policies $\mathcal{X} \to \mathcal{P}(\mathcal{A})$, against which we would like our policy to compete; but the adaptive policies by Badanidiyuru et al. [2014] and Agrawal et al. [2016] only compete with respect to the best single element in $\Pi$, not the best convex combination of elements of $\Pi$.

The setting of linear CBwK (Agrawal and Devanur [2016]) provides a structural link between contexts and expected rewards and cost vectors, but in a linear way that is incomparable to the setting of CBwK for a conversion model introduced above. More details are given in Section 6. We also mention that linear and logistic structural links between contexts (prices) and rewards or costs were studied in a non-contextual setting (i.e., not in CBwK) by Miao et al. [2021]. Their strategy bears some resemblance to the one by Agrawal and Devanur [2016], in particular, both consider an online convex optimization strategy as a subroutine.

All mentioned references consider a no-op action $a_{\text{null}}$. (It could be replaced by the existence of a standard action $a_{\text{no-cost}}$ always achieving null costs and possibly some positive rewards.)

On the contrary, none of the mentioned references assumes that the context $\mathcal{X}$ set is finite. This is a technical necessity for a part of the adaptive policy introduced; see the discussion of computational complexity at the end of Section 3. But somehow, considering a finite set $\Pi$ of policies, as in Badanidiyuru et al. [2014] and Agrawal et al. [2016], is a counterpart to assuming finiteness of $\mathcal{X}$. Also, Appendix F actually mitigates this restriction that $\mathcal{X}$ is finite: learning the logistic parameter $\boldsymbol{\theta}_\star$ may be achieved with continuous contexts (see Phase 1 in Section 3); only the subsequent

optimization part (Phase 2 in Section 3) requires finiteness of $\mathcal{X}$. We may thus well discretize only $\mathcal{X}$ for this Phase 2, which is exactly what Appendix F performs. This mitigation comes with possible theoretical guarantees as Sections 4 and 5 reveal that the errors $\varepsilon_t(a, \boldsymbol{x})$ for learning $\theta_\star$ and $P$, obtained as outcomes of the first step of the analyses, are carried over in the subsequent steps, where the optimization part is evaluated.

## 3 Description of the Adaptive Policy Considered

At each stage $t \geqslant 1$, the policy first updates an estimator $\widehat{\boldsymbol{\theta}}_{t-1}$ of $\boldsymbol{\theta}_\star$ based on the history $h_{t-1}$ available so far, based on an adaptation of the Logistic-UCB1 algorithm by Faury et al. [2020], and deduces estimators $\widehat{P}_{t-1}(a, \boldsymbol{x})$ and upper confidence bounds $U_{t-1}(a, \boldsymbol{x})$ of the probabilities $P(a, \boldsymbol{x})$. The policy then solves the corresponding estimated version of the optimization problem (2). We now describe the corresponding two steps. In the description below, quantities that depend on information available at round $t - 1$ (respectively, $t$) are indexed by $t - 1$ (respectively, $t$).

**Phase 0: In case the cost constraints are about to be violated.** To make sure cost constraints are never violated, whenever at least one of the components of the current cumulative costs is larger than $B - 1$ and could possibly be larger than $B$ at the end of round $t$, we play $a_{\text{null}}$ (and we actually do so for the rest of the rounds). This corresponds to defining $\boldsymbol{p}_t(h_{t-1}, \boldsymbol{x}) = \delta_{a_{\text{null}}}$ for all $\boldsymbol{x} \in \mathcal{X}$, where $\delta_{a_{\text{null}}}$ denotes the Dirac mass on $a_{\text{null}}$. Otherwise, we proceed as described below in Phase 1 and Phase 2.

**Phase 1: Learning $\theta_\star$ via an adapted Logistic-UCB1.** This first phase depends on a regularization parameter $\lambda > 0$ and on upper-confidence bonuses $\varepsilon_t(a, \boldsymbol{x}) > 0$, both to be specified by the analysis.

At rounds $t \geqslant 2$, we first maximize a regularized log-likelihood of the history $h_{t-1}$:

$$\tilde{\boldsymbol{\theta}}_{t-1} \in \underset{\boldsymbol{\theta} \in \mathbb{R}^m}{\operatorname{argmax}} \sum_{s=1}^{t-1} \mathbb{1}_{\{a_s \neq a_{\text{null}}\}} \left( y_s \ln \eta\big(\boldsymbol{\varphi}(a_s, \boldsymbol{x}_s)^{\mathrm{T}} \boldsymbol{\theta}\big) + (1 - y_s) \ln\Big(1 - \eta\big(\boldsymbol{\varphi}(a_s, \boldsymbol{x}_s)^{\mathrm{T}} \boldsymbol{\theta}\big)\Big) \right) - \frac{\lambda}{2}\|\boldsymbol{\theta}\|^2 . \tag{3}$$

In the expression above, we read that we only gather information about $\boldsymbol{\theta}_\star$ at those rounds $s$ when $a_s \neq a_{\text{null}}$. When $\tilde{\boldsymbol{\theta}}_{t-1}$ does not belong to $\Theta$, an ad hoc projection step corrects for this, if needed:

$$\widehat{\boldsymbol{\theta}}_{t-1} \in \underset{\boldsymbol{\theta} \in \Theta}{\operatorname{argmin}} \left\| \Psi_{t-1}(\boldsymbol{\theta}) - \Psi_{t-1}(\tilde{\boldsymbol{\theta}}_{t-1}) \right\|_{W_{t-1}(\boldsymbol{\theta})^{-1}}, \tag{4}$$

$$\text{where} \qquad \Psi_{t-1}(\boldsymbol{\theta}) = \sum_{s=1}^{t-1} \mathbb{1}_{\{a_s \neq a_{\text{null}}\}} \, \eta\big(\boldsymbol{\varphi}(a_s, \boldsymbol{x}_s)^{\mathrm{T}} \boldsymbol{\theta}\big) \, \boldsymbol{\varphi}(a_s, \boldsymbol{x}_s) + \lambda \boldsymbol{\theta}$$

$$\text{and} \qquad W_{t-1}(\boldsymbol{\theta}) = \lambda \, \mathrm{I}_m + \sum_{s=1}^{t-1} \mathbb{1}_{\{a_s \neq a_{\text{null}}\}} \, \dot{\eta}\big(\boldsymbol{\varphi}(a_s, \boldsymbol{x}_s)^{\mathrm{T}} \boldsymbol{\theta}\big) \, \boldsymbol{\varphi}(a_s, \boldsymbol{x}_s) \boldsymbol{\varphi}(a_s, \boldsymbol{x}_s)^{\mathrm{T}} . \tag{5}$$

We recall that the function $\dot{\eta}$ denotes the derivative of $\eta$, i.e., $\dot{\eta}(x) = \mathrm{e}^{-x} / (1 + \mathrm{e}^{-x})^2$. We have $\dot{\eta} = \eta(1 - \eta)$.

By plug-in, we finally define estimators and upper-confidence bounds of the probabilities $P(a, \boldsymbol{x})$ for $a \neq a_{\text{null}}$ and all $\boldsymbol{x} \in \mathcal{X}$:

$$\widehat{P}_{t-1}(a, \boldsymbol{x}) = \eta\big(\boldsymbol{\varphi}(a, \boldsymbol{x})^{\mathrm{T}} \widehat{\boldsymbol{\theta}}_{t-1}\big) \qquad \text{and} \qquad U_{t-1}(a, \boldsymbol{x}) = \min\big\{\widehat{P}_{t-1}(a, \boldsymbol{x}) + \varepsilon_{t-1}(a, \boldsymbol{x}), \, 1\big\} .$$

For $a_{\text{null}}$, no estimators or upper-confidence bounds need to be defined, as the quantities $P(a_{\text{null}}, \boldsymbol{x})$ are actually undefined.

**Phase 2: Sampling, via solving an optimization problem with expected constraints.** This phase relies on a conservative-budget parameter denoted by $B_T$, which is only slightly smaller than $B$ and whose exact value is to be specified by the analysis.

We start with the case of a known context distribution $\nu$. At round $t = 1$, we play an arbitrary action in $\mathcal{A} \setminus \{a_{\text{null}}\}$. At rounds $t \geqslant 2$, if at least one component of the cumulative vector costs suffered so far is larger than $B - 1$, we pick $a_t = a_{\text{null}}$. Otherwise, we pick for $\boldsymbol{p}_t(h_{t-1}, \cdot)$ the solution of the optimization problem $\operatorname{OPT}(\nu, U_{t-1}, B_T)$ defined[1] in (2), and draw $a_t$ according to $\boldsymbol{p}_t(h_{t-1}, \boldsymbol{x}_t)$.

---

[1]In the definition (2) of $\operatorname{OPT}(\nu, U_{t-1}, B_T)$, expectations are only over $\boldsymbol{X} \sim \nu$ and not over the random variable $U_{t-1}$; more comments and explanations on this fact may be found in Appendix B.3.



BOX B: LOGISTIC-UCB1 FOR DIRECT SOLUTIONS TO OPT PROBLEMS

**Parameters:** regularization parameter $\lambda > 0$; conservative-budget parameter $B_T$; upper-confidence bonuses $\varepsilon_s(a, \boldsymbol{x}) > 0$, for $s \geqslant 1$ and $(a, \boldsymbol{x}) \in \big(\mathcal{A} \setminus \{a_{\text{null}}\}\big) \times \mathcal{X}$.

**Round** $t = 1$: play an arbitrary action $a_1 \in \mathcal{A} \setminus \{a_{\text{null}}\}$

**At rounds** $t \geqslant 2$:

Phase 0 If $\displaystyle\sum_{s \leqslant t-1} \boldsymbol{c}(a_s, \boldsymbol{x}_s)\, y_s \leqslant (B-1)\mathbf{1}$ is violated, then $\boldsymbol{p}_t(h_{t-1}, \boldsymbol{x}) = \delta_{a_{\text{null}}}$ for all $\boldsymbol{x}$

Phase 1 Otherwise, compute a maximum-likelihood estimator $\tilde{\boldsymbol{\theta}}_{t-1}$ of $\boldsymbol{\theta}_\star$ according to (3), compute its projection $\widehat{\boldsymbol{\theta}}_{t-1}$ onto $\Theta$ according to (4), and define, for $a \neq a_{\text{null}}$:

$$\widehat{P}_{t-1}(a, \boldsymbol{x}) = \eta\big(\boldsymbol{\varphi}(a, \boldsymbol{x})^{\mathsf{T}} \widehat{\boldsymbol{\theta}}_{t-1}\big) \quad \text{and} \quad U_{t-1}(a, \boldsymbol{x}) = \min\Big\{\widehat{P}_{t-1}(a, \boldsymbol{x}) + \varepsilon_{t-1}(a, \boldsymbol{x}),\ 1\Big\}$$

Phase 2 Compute the solution $\boldsymbol{p}_t(h_{t-1}, \cdot)$ of

$$\text{OPT}\big(\tilde{\nu}, U_{t-1}, B_T\big) = \max_{\pi:\mathcal{X} \to \mathcal{P}(\mathcal{A})} T\, \mathbb{E}_{\boldsymbol{X} \sim \tilde{\nu}}\Bigg[\sum_{a \in \mathcal{A}} r(a, \boldsymbol{X})\, U_{t-1}(a, \boldsymbol{X})\, \pi_a(\boldsymbol{X})\Bigg]$$

$$\text{under} \qquad T\, \mathbb{E}_{\boldsymbol{X} \sim \tilde{\nu}}\Bigg[\sum_{a \in \mathcal{A}} \boldsymbol{c}(a, \boldsymbol{X})\, U_{t-1}(a, \boldsymbol{X})\, \pi_a(\boldsymbol{X})\Bigg] \leqslant B_T \mathbf{1}\,,$$

where $\tilde{\nu}$ denotes either $\nu$ (when it is known) or its empirical estimate $\widehat{\nu}_t$ in (6)
Draw an arm $a_t \sim \boldsymbol{p}_t(h_{t-1}, \boldsymbol{x}_t)$.



When the context distribution is unknown, we rather pick for $\boldsymbol{p}_t(h_{t-1}, \cdot)$ the solution of the optimization problem $\text{OPT}\big(\widehat{\nu}_t, U_{t-1}, B_T\big)$, where

$$\widehat{\nu}_t = \frac{1}{t} \sum_{s=1}^{t} \delta_{\boldsymbol{x}_s}\,, \tag{6}$$

with $\delta_{\boldsymbol{x}}$ denoting the Dirac mass at $\boldsymbol{x} \in \mathcal{X}$. Since $\boldsymbol{x}_t$ is revealed at the beginning of round $t$, before we pick an action, we may indeed use $\widehat{\nu}_t$ at round $t$.

**Summary and discussion of the computational complexity.** We summarize the considered adaptive policy in Box B and now discuss its computational complexity.

As $\ln \varphi$ and $\ln(1 - \varphi)$ are strictly concave and smooth, the maximum-likelihood step (3) of Phase 1 consists of maximizing a strictly concave and smooth function over $\mathbb{R}^m$, which may be performed efficiently. The projection step (4) of Phase 1 is however an issue, both with the version of Logistic-UCB1 discussed here and with the earlier approach by Filippi et al. [2010, Section 3]. The latter and Faury et al. [2020, Section 4.1] both underline that the projection step (4) is a complex optimization problem that however does not often need to be solved in practice, as they usually observe $\tilde{\boldsymbol{\theta}}_{t-1} \in \Theta$. Our numerical experiments concur with this statement (but admittedly, they rely on choosing a rather large value of $\Theta$).

On the contrary, Phase 2 of the adaptive policy consists of solving a linear program with $|\mathcal{X}| \times |\mathcal{A}|$ constraints, where where $|\mathcal{X}|$ and $\mathcal{A}$ denote the cardinality of $\mathcal{X}$ and $\mathcal{A}$, respectively—see the detailed rewriting (13) in the supplementary material. Therefore, the computational complexity of Phase 2 is polynomial (of weak order) in $|\mathcal{X}| \times |\mathcal{A}|$. To achieve this acceptable complexity we had however to restrict our attention to finite sets of contexts $\mathcal{X}$, which requires in practice segmenting countable or continuous context sets into finitely many clusters, for instance. We do so in our experiments.

**Simulation study.** A simulation study on partially simulated but realistic data may be found in Appendix F. The underlying dataset is the standard "default of credit card clients" dataset of UCI [2016], initially provided by Yeh and Lien [2009]. (It may be used under a Creative Commons Attribution 4.0 International [CC BY 4.0] license.)

# 4 Analysis for a Known Context Distribution $\nu$

Since $\Theta$ is bounded, the following quantity, standardly introduced in the context of logistic bandits (see Faury et al. [2020] and references therein), is finite, though possibly large:

$$\kappa = \sup\left\{\frac{1}{\dot{\eta}(\boldsymbol{\varphi}(a, \boldsymbol{x})^{\mathrm{T}} \boldsymbol{\theta})} : \boldsymbol{x} \in \mathcal{X}, \ a \in \mathcal{A} \setminus \{a_{\mathrm{null}}\}, \ \boldsymbol{\theta} \in \Theta\right\} < +\infty \,.$$

We denote by $\|\Theta\| = \max\{\|\boldsymbol{\theta}\| : \boldsymbol{\theta} \in \Theta\}$ the maximal Euclidean norm of an element in $\Theta$.

By construction, given that individual cost vectors lie in $[0, 1]^d$ and due to its "Phase 0", the adaptive policy considered always satisfies the budget constraints. The bound on rewards reads as follows.

**Theorem 1.** *In the setting of Box A of Section 2.1, we consider the adaptive policy of Box B of Section 3 assuming that the distribution of the contexts is known, i.e., with $\tilde{\nu} = \nu$. We set a confidence level $1 - \delta \in (0, 1)$ and use parameters $\lambda = m \ln(1 + T/m)$,*

$$B_T = B - 2 - \sqrt{2T \ln(4d/\delta)} \,,$$

*and $\varepsilon_t(a, \boldsymbol{x})$ stated in (9) of the supplementary material. Then, provided that $T \geqslant 2m$ and $B > 4 + 2\sqrt{2T \ln(4d/\delta)}$, we have, with probability at least $1 - 2\delta$,*

$$\mathrm{OPT}(\nu, P, B) - \sum_{t \leqslant T} r(a_t, \boldsymbol{x}_t) \, y_t \leqslant \left(4 + 2\sqrt{2T \ln \frac{4d}{\delta}}\right) \frac{\mathrm{OPT}(\nu, P, B)}{B} + E_T + \sqrt{2T \ln \frac{4}{\delta}} + 1 \,,$$

*where the closed-form expression of $E_T = \mathcal{O}\big(m\sqrt{T} \ln T\big)$ is in (35) of the supplementary material.*

We will rather discuss the bound of the more general Theorem 2 (to be stated and proved in Section 5) than the one of Theorem 1. We provide a proof sketch in Section 4.1 and discuss the main technical novelty in Section 4.2.

## 4.1 Proof Sketch for Theorem 1

The detailed proof of Theorem 1 may be found in Appendix B. We provide here an overview thereof, highlighting the four main ingredients. The third and fourth steps benefited from some inspiration drawn from the proof techniques of Agrawal and Devanur [2016]. The first step is an adaptation of Lemmas 1 and 2 by Faury et al. [2020].

**First,** the mentioned adaptation provides values of the parameters $\varepsilon_t(a, \boldsymbol{x})$ such that, with probability at least $1 - \delta$,

$$\forall t \geqslant 1, \ \forall a \in \mathcal{A} \setminus \{a_{\mathrm{null}}\}, \ \forall \boldsymbol{x} \in \mathcal{X}, \qquad \left|\widehat{P}_t(a, \boldsymbol{x}) - P(a, \boldsymbol{x})\right| \leqslant \varepsilon_t(a, \boldsymbol{x}) \,,$$

hence
$$U_t(a, \boldsymbol{x}) - 2\varepsilon_t(a, \boldsymbol{x}) \leqslant P(a, \boldsymbol{x}) \leqslant U_t(a, \boldsymbol{x}) \,,$$

while $\displaystyle\sum_{t \leqslant T} \varepsilon_{t-1}(a_t, \boldsymbol{x}_t) \mathbb{1}_{\{a_t \neq a_{\mathrm{null}}\}}$ is of order $\sqrt{T}$ up to poly-logarithmic terms.

**Second,** the Phase 2 formulation of the strategy, in a primal form, is equivalently restated in a dual form. For each round $t \geqslant 2$, strong duality holds and entails the existence of a vector $\boldsymbol{\beta}_t^{\mathrm{budg},\star} \in \mathbb{R}^d$ such that $\boldsymbol{p}_t(h_{t-1}, \cdot)$ may be identified as the argmax over $\pi : \mathcal{X} \to \mathcal{P}(\mathcal{A})$ of

$$\mathbb{E}_{\boldsymbol{X} \sim \nu}\left[T \sum_{a \in \mathcal{A}} \left(r(a, \boldsymbol{X}) - \left(\boldsymbol{\beta}_t^{\mathrm{budg},\star}\right)^{\mathrm{T}} \boldsymbol{c}(a, \boldsymbol{X})\right) U_{t-1}(a, \boldsymbol{X}) \, \pi_a(\boldsymbol{X}) + \sum_{\boldsymbol{x} \in \mathcal{X}} \sum_{a \in \mathcal{A}} \beta_{\boldsymbol{x},a}^{\mathrm{p\text{-}pos},\star} \, \pi_a(\boldsymbol{x})\right] \,.$$

By exploiting the KKT conditions, we are able to get rid of the double sum above and finally get a $\mathcal{X}$–pointwise characterization of $\boldsymbol{p}_t(h_{t-1}, \cdot)$: for all $\boldsymbol{x} \in \mathcal{X}$,

$$\boldsymbol{p}_t(h_{t-1}, \boldsymbol{x}) \in \operatorname*{argmax}_{\boldsymbol{q} \in \mathcal{P}(\mathcal{A})} \sum_{a \in \mathcal{A}} \left(r(a, \boldsymbol{x}) - \left(\boldsymbol{\beta}_t^{\mathrm{budg},\star}\right)^{\mathrm{T}} \boldsymbol{c}(a, \boldsymbol{x})\right) U_{t-1}(a, \boldsymbol{x}) \, q_a$$

$$= \operatorname*{argmax}_{\boldsymbol{q} \in \mathcal{P}(\mathcal{A})} \sum_{a \in \mathcal{A}} \left(r(a, \boldsymbol{x}) - \left(\boldsymbol{\beta}_t^{\mathrm{budg},\star}\right)^{\mathrm{T}} \boldsymbol{c}(a, \boldsymbol{x})\right)_+ U_{t-1}(a, \boldsymbol{x}) \, q_a \,.$$

Non-negative parts $(\cdot)_+$ may be introduced thanks to the existence of the no-op action $a_{\text{null}}$. The distributions $\boldsymbol{p}_t(h_{t-1}, \boldsymbol{x})$ may therefore be interpreted as maximizing some upper-confidence bound on penalized gains (rewards minus some scalarized costs); the dual variables $\boldsymbol{\beta}_t^{\text{budg},\star}$ play a role similar to the $Z$ parameter of Agrawal and Devanur [2016, Section 3.3] in terms of weighing gains versus costs. In passing, we also prove

$$\text{OPT}(\nu, U_{t-1}, B_T) \geqslant B_T (\boldsymbol{\beta}_t^{\text{budg},\star})^{\mathrm{T}} \mathbf{1}$$

based on the KKT conditions. The latter inequality is comparable in spirit to the bound of Agrawal and Devanur [2016, Corollary 3], relating $Z$ to $\text{OPT}(\nu, P, B)/B$.

**Third,** for $t \geqslant 2$, whenever the policy $\boldsymbol{p}_t(h_{t-1}, \cdot)$ is obtained by solving the optimization problem $\text{OPT}(\nu, U_{t-1}, B_T)$ of Phase 2 and by independence of $\boldsymbol{x}_t$ and $h_{t-1}$, we have

$$\frac{\text{OPT}(\nu, U_{t-1}, B_T)}{T} = \mathbb{E}_{\boldsymbol{X} \sim \nu} \left[ \sum_{a \in \mathcal{A}} r(a, \boldsymbol{X}) \, U_{t-1}(a, \boldsymbol{X}) \, p_{t,a}(h_{t-1}, \boldsymbol{X}) \right]$$

$$= \mathbb{E} \big[ r(a_t, \boldsymbol{x}_t) \, U_{t-1}(a_t, \boldsymbol{x}_t) \, \big| \, h_{t-1} \big] .$$

Therefore, repeated applications of the Hoeffding-Azuma inequality and the inequalities of the first step entail that, up to quantities of the order of $\sqrt{T}$,

$$\sum_{t=2}^{T} \frac{\text{OPT}(\nu, U_{t-1}, B_T)}{T} \approx \sum_{t=2}^{T} r(a_t, \boldsymbol{x}_t) U_{t-1}(a_t, \boldsymbol{x}_t)$$

$$\lesssim \sum_{t=2}^{T} \varepsilon_{t-1}(a_t, \boldsymbol{x}_t) \mathbb{1}_{\{a_t \neq a_{\text{null}}\}} + \sum_{t=2}^{T} r(a_t, \boldsymbol{x}_t) P(a_t, \boldsymbol{x}_t) \lesssim \sum_{t=2}^{T} r(a_t, \boldsymbol{x}_t) \, y_t .$$

We thus only need to control $\text{OPT}(\nu, P, B) - \sum_{t=2}^{T} \dfrac{\text{OPT}(\nu, U_{t-1}, B_T)}{T}$, which may be assumed $\geqslant 0$.

The value $B_T = B - 2 - \sqrt{2T \ln(4d/\delta)}$ and similar Hoeffding-Azuma-based arguments show that with high probability, the budget limit $B - 1$ is indeed never reached and that we always compute $\boldsymbol{p}_t(h_{t-1}, \cdot)$ in the way indicated by Phase 2.

**Fourth,** we collect all bounds together. We start with

$$\sum_{t=2}^{T} \frac{B_T}{T} (\boldsymbol{\beta}_t^{\text{budg},\star})^{\mathrm{T}} \mathbf{1} \leqslant \sum_{t=2}^{T} \frac{\text{OPT}(\nu, U_{t-1}, B_T)}{T} \leqslant \text{OPT}(\nu, P, B) .$$

We the exploit the dual characterization of $\boldsymbol{p}_t(h_{t-1}, \cdot)$ and the control $P \leqslant U_{t-1}$ to get that with high probability, for all $\boldsymbol{x} \in \mathcal{X}$,

$$\sum_{a \in \mathcal{A}} \big( r(a, \boldsymbol{x}) - (\boldsymbol{\beta}_t^{\text{budg},\star})^{\mathrm{T}} \boldsymbol{c}(a, \boldsymbol{x}) \big) U_{t-1}(a, \boldsymbol{x}) \, p_{t,a}(h_{t-1}, \boldsymbol{x})$$

$$\geqslant \sum_{a \in \mathcal{A}} \big( r(a, \boldsymbol{x}) - (\boldsymbol{\beta}_t^{\text{budg},\star})^{\mathrm{T}} \boldsymbol{c}(a, \boldsymbol{x}) \big) P(a, \boldsymbol{x}) \, \pi_a^\star(\boldsymbol{x}) .$$

After integration over $\boldsymbol{X} \sim \nu$ and substituting of the definitions of $\pi^\star$ and $\boldsymbol{p}_{t,a}(h_{t-1}, \cdot)$, as well as the equality stemming from the KKT conditions, we have

$$\overbrace{\mathbb{E}_{\boldsymbol{X} \sim \nu} \left[ \sum_{a \in \mathcal{A}} r(a, \boldsymbol{X}) \, U_{t-1}(a, \boldsymbol{X}) \, \boldsymbol{p}_{t,a}(h_{t-1}, \boldsymbol{X}) \right]}^{= \text{OPT}(\nu, U_{t-1}, B_T)/T}$$

$$\underbrace{- \mathbb{E}_{\boldsymbol{X} \sim \nu} \left[ \sum_{a \in \mathcal{A}} (\boldsymbol{\beta}_t^{\text{budg},\star})^{\mathrm{T}} \boldsymbol{c}(a, \boldsymbol{X}) \, U_{t-1}(a, \boldsymbol{X}) \, \boldsymbol{p}_{t,a}(h_{t-1}, \boldsymbol{X}) \right]}_{(B_T/T)(\boldsymbol{\beta}_t^{\text{budg},\star})^{\mathrm{T}} \mathbf{1}}$$

$$\geqslant \underbrace{\mathbb{E}_{\boldsymbol{X} \sim \nu} \left[ \sum_{a \in \mathcal{A}} r(a, \boldsymbol{X}) \, P(a, \boldsymbol{X}) \, \pi_a^\star(\boldsymbol{X}) \right]}_{= \text{OPT}(\nu, P, B)/T} - (\boldsymbol{\beta}_t^{\text{budg},\star})^{\mathrm{T}} \underbrace{\mathbb{E}_{\boldsymbol{X} \sim \nu} \left[ \sum_{a \in \mathcal{A}} \boldsymbol{c}(a, \boldsymbol{X}) \, P(a, \boldsymbol{X}) \, \pi_a^\star(\boldsymbol{X}) \right]}_{\leqslant (B/T)\mathbf{1}} .$$

Rearranging and summing over $2 \leqslant t \leqslant T$, we obtain

$$\sum_{t=2}^{T} \frac{\text{OPT}(\nu, P, B) - \text{OPT}(\nu, U_{t-1}, B_T)}{T} \leqslant \sum_{t=2}^{T} \frac{B - B_T}{T} (\boldsymbol{\beta}_t^{\text{budg},\star})^{\text{T}} \mathbf{1} \leqslant \left( \frac{B}{B_T} - 1 \right) \text{OPT}(\nu, P, B),$$

where we substituted the first inequality stated in this fourth step. This concludes the proof.

## 4.2 Discussion on the Main Technical Novelties

As should be clear from the comments at the beginning of Section 4.1, the technical novelties lies in the second step of the proof of Theorem 1. On the one hand, we are able to directly analyze a strategy stated in a primal form, which is a more natural formulation. On the other hand, doing so, we are also able to avoid the issues that come with dual formulations, relying, e.g., on some critical parameter $Z$, as in Agrawal and Devanur [2016, Theorem 3], to trade off rewards and costs. This parameter $Z$ should be of order $\text{OPT}/B$ and has to be learned, e.g., through $\sqrt{T}$ initial exploration rounds. (More details are to be found in Section E.3.) In our analysis, this parameter $Z$ is superseded by dual optimal variables $\boldsymbol{\beta}_t^{\text{budg},\star} \geqslant \mathbf{0}$, that are only used in the analysis and not to state the policy, unlike in Agrawal and Devanur [2016]. Put differently, the clever use in this context of KKT conditions is the main technical novelty. On a side note, we are also able to take care in an explicit and detailed fashion of the no-op action $a_{\text{null}}$, whose specific treatment is often unaddressed in the literature.

## 5 Analysis for an Unknown Context Distribution $\nu$

When the context distribution $\nu$ is unknown, we simply estimate it through its empirical frequencies (6). The regret bound is almost unchanged: an additional mild factor of, e.g., $2|\mathcal{X}|\sqrt{2T \ln(2T|\mathcal{X}|/\delta)}$ appears in the $\sqrt{T}$ term multiplying $\text{OPT}(\nu, P, B)/B$. This term comes from some uniform deviation argument stated in (7) and 8.

**Theorem 2.** *In the setting of Box A of Section 2.1, we consider the adaptive policy of Box B of Section 3 with $\tilde{\nu} = \widehat{\nu}_t$ at rounds $t \geqslant 2$. We set a confidence level $1 - \delta \in (0, 1)$ and use parameters $\lambda = m \ln(1 + T/m)$, a working budget of*

$$B - b_T, \qquad \text{where} \qquad b_T = 2 + \sqrt{2T \ln(4d/\delta)} + |\mathcal{X}|\sqrt{2T \ln(2T|\mathcal{X}|/\delta)},$$

*and $\varepsilon_t(a, \boldsymbol{x})$ stated in (9) of the supplementary material. Then, provided that $T \geqslant 2m$ and $B > 2b_T$, we have, with probability at least $1 - 3\delta$,*

$$\text{OPT}(\nu, P, B) - \sum_{t \leqslant T} r(a_t, \boldsymbol{x}_t) \, y_t \leqslant 2b_T \left( 1 + \frac{\text{OPT}(\nu, P, B)}{B} \right) + E_T,$$

*where the expression of $E_T = \mathcal{O}(m\sqrt{T} \ln T)$ may be found in (35) of the supplementary material.*

The order of magnitude of the regret bound is $(m + |\mathcal{X}|\text{OPT}(\nu, P, B)/B)\sqrt{T} \ln T$, which is reminiscent of all known regret upper bounds for CBwK (e.g., the ones by Badanidiyuru et al. [2014] and Agrawal et al. [2016], for general CBwK, and Agrawal and Devanur [2016] for linear CBwK, see Section 6). The factor $|\mathcal{X}|$ may be improved, see below, but this is a detail. A discussion on exhibiting corresponding lower bounds is to be found at the end of Section 6.

A detailed proof of Theorem 2 is provided in Appendix D of the supplementary material. It follows closely the proof of Theorem 1, with modifications mostly consisting of relating quantities of the form

$$\mathbb{E}_{\boldsymbol{X} \sim \widehat{\nu}_t} \big[ f(\boldsymbol{X}) \big] \quad \text{vs.} \quad \mathbb{E}_{\boldsymbol{X} \sim \nu} \big[ f(\boldsymbol{X}) \big], \quad \text{where, e.g.,} \quad f(\boldsymbol{X}) = \sum_{a \in \mathcal{A}} r(a, \boldsymbol{X}) \, U_{t-1}(a, \boldsymbol{X}) \, \boldsymbol{p}_{t,a}(h_{t-1}, \boldsymbol{X}).$$

To do so, we use that for all functions $f : \mathcal{X} \to [0, 1]$,

$$\forall t \leqslant T, \qquad \left| \mathbb{E}_{\boldsymbol{X} \sim \widehat{\nu}_t} \big[ f(\boldsymbol{X}) \big] - \mathbb{E}_{\boldsymbol{X} \sim \nu} \big[ f(\boldsymbol{X}) \big] \right| \leqslant \sum_{x \in \mathcal{X}} \big| \widehat{\nu}_t(x) - \nu(x) \big| \stackrel{\text{def}}{=} \big\| \widehat{\nu}_t - \nu \big\|_1, \quad (7)$$

where $\big\| \widehat{\nu}_t - \nu \big\|_1$ is the total variation distance between $\widehat{\nu}_t$ and $\nu$. In Appendix D, we upper bound the latter, for the sake of simplicity, in a crude way by applying $T|\mathcal{X}|$ times the Hoeffding-Azuma inequality (once for each $1 \leqslant t \leqslant T$ and $\boldsymbol{x} \in \mathcal{X}$) and obtain that with probability at least $1 - \delta$,

$$\forall t \leqslant T, \qquad \big\| \widehat{\nu}_t - \nu \big\|_1 \leqslant |\mathcal{X}|\sqrt{\frac{1}{2t} \ln \frac{2T|\mathcal{X}|}{\delta}}. \quad (8)$$

The $|\mathcal{X}|\sqrt{2T\ln(2T|\mathcal{X}|/\delta)}$ term in the regret bound of Theorem 2 appears as the sum over $t \leqslant T$ of the deviation bounds (8). The bounds (8) may actually be improved into bounds of the order of $\sqrt{|\mathcal{X}|/t}$, via some Cauchy-Schwarz bound and a deviation argument in Banach spaces by Pinelis [1994], or by more direct techniques described by Devroye [1983, Lemma 3] and Berend and Kontorovich [2012]. In any case, the regret bound of Theorem 2 automatically benefits from such improvements, by replacing the $2|\mathcal{X}|\sqrt{2T\ln(2T|\mathcal{X}|/\delta)}$ term in the bound therein by the (sum over $t \leqslant T$ of the) better uniform deviation bounds.

## 6 Extension to Linear Contextual Bandits with Knapsacks

This section is a brief summary of Appendix E. We explain therein how the adaptive policy of Box B may be adapted to the setting of linear CBwK, introduced by Agrawal and Devanur [2016], where the bounded rewards $r_t$ and vector costs $\boldsymbol{c}_t$ are independently generated at each round according to bounded distributions with respective expectations $\overline{r}(a_t, \boldsymbol{x}_t)$ and $\overline{\boldsymbol{c}}(a_t, \boldsymbol{x}_t)$, depending linearly on (a transfer function $\varphi$ of) the contexts: for all $a \neq a_{\text{null}}$ and $\boldsymbol{x} \in \mathcal{X}$, for all components $i$ of $\overline{\boldsymbol{c}}$,

$$\overline{r}(a, \boldsymbol{x}) = \boldsymbol{\varphi}(a, \boldsymbol{x})^{\mathrm{T}} \boldsymbol{\mu}_\star \qquad \text{and} \qquad \overline{c}_i(a, \boldsymbol{x}) = \boldsymbol{\varphi}(a, \boldsymbol{x})^{\mathrm{T}} \boldsymbol{\theta}_{\star, i} \,.$$

We consider the same benchmark $\text{OPT}(\nu, \overline{r}, \overline{\boldsymbol{c}}, B)$ as Agrawal and Devanur [2016] and are able to exhibit a similar $\big(\text{OPT}(\nu, \overline{r}, \overline{\boldsymbol{c}}, B)/B\big)m\sqrt{T}\ln T$ regret bound, with however a slight relaxation on the order of magnitude required for $B$. We do so with a strategy that we deem more direct and natural, inspired from the one of Box B, where in Phase 1 a LinUCB-type (Abbasi-Yadkori et al. [2011]) estimation of the parameters is performed, and where in Phase 2, a direct solution to an $\text{OPT}$ problem with estimated parameters is performed. The parameters are upper-confidence functions $U_{t-1}$ on $\overline{r}$ and lower-confidence vector functions $\boldsymbol{L}_{t-1}$ on $\overline{\boldsymbol{c}}$.

The main advantage of our approach in the case of linear contextual bandits is exactly as described in Section 4.2: avoiding the critical parameter $Z$ of Agrawal and Devanur [2016, Theorem 3], which is used to trade off rewards and costs. The main limitation of our approach is the assumption of a finite context set $\mathcal{X}$, which is required to make the Phase-2 linear program tractable.

## 7 Future Work

We conclude this article with a list of issues to be further investigated.

*First*, as discussed in Section 2.2, the restrictions of finiteness should be alleviated: finiteness of the context set in the setting of this article, or finiteness of the set of benchmark policies in other settings (see Badanidiyuru et al., 2014 and Agrawal et al., 2016).

*Second*, we only dealt with $\leqslant$ budget constraints (and do does the literature so far). Direct approaches to constraints of the form $\geqslant$ remain to be further investigated.

*A third* series of questions to be clarifies concerns regret lower bounds, and more generally, the tightness of the results—in particular, the required conditions on budget sizes. Earlier references for contextual bandits with knapsacks did also not provide lower bounds statements that were simultaneously optimal (i.e., matching the obtained upper bounds) and general (i.e., valid for all problems with a given number of rounds $T$, a given budget $B$, and a given value $\text{OPT}$ for the optimal expected reward achievable by a static policy). Badanidiyuru et al. [2014, comments after Theorem 1] merely indicates that the obtained regret upper bound is optimal in some regimes, e.g., when the budget $B$ grows linearly with the number of rounds $T$. Agrawal and Devanur [2016, comments after Theorem 1] only compares the obtained regret upper bound to the case of no budget constraints. In particular, as far as the orders of magnitude in $T$ are concerned, it is unclear whether the $(\text{OPT}/B)\sqrt{T}$ rates achieved (up to poly-logarithmic factors) in the present article and in the two mentioned references are optimal. These rates do not match the optimal rates in the case of no contexts, which were stated and proved by Badanidiyuru et al. [2013].

## Acknowledgments and Disclosure of Funding

Zhen Li and Gilles Stoltz have no direct funding to acknowledge other than the salaries paid by their employers, BNP Paribas, CNRS, and HEC Paris. They have no competing interests to declare.

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
