# OpenReview forum: "Contextual Bandits with Knapsacks for a Conversion Model"
_NeurIPS.cc/2022/Conference — NeurIPS 2022 Accept_

### Official Review · Reviewer_xxh2 · 2022-07-10

**Rating:** 7
**Confidence:** 3
**Soundness:** 4 excellent
**Presentation:** 4 excellent
**Contribution:** 3 good

**Summary:**

This paper proposes a new contextual bandit model with knapsacks. In this new model, the decision-maker sequentially offers actions to arriving users after observing their contexts. Rewards and costs are only realized if the users accept the offer (a conversion happens). Conversion probabilities follow a logistic model that depends on action-context features and an unknown parameter vector. The goal is to design an adaptive policy that performs almost as well as the best static policy that knows the context distribution and conversion probabilities. The authors propose logistic-UCB1 algorithm to solve this problem. The proposed algorithm at each round computes the MLE estimate of the unknown parameter vector based on history. Then, it computes action selection probabilities by solving an optimization problem with expected constraints. This is a linear program that can be computationally manageable for finite context spaces. Sublinear regret upper bounds are shown both when the learner knows the context distribution and when the learner uses an empirical estimate of the context distribution. Supplemental material provides a motivating example from the banking sector, where the actions correspond to discounts on offered loans. They also investigate how to extend the current work to linear contextual bandits with knapsacks and come up with an algorithm with better budget requirements and parameter tuning capabilities, albeit requiring the finiteness of contexts.

**Questions:**

Why negative regret appears in simulations?

When justifying comparing with the optimal static policy in regret, the work of Agrawal and Devanur (2016) is mentioned. However, the setup here is not the same due to the conversion model. How is it ensured that the relation between the optimal static policy and the best feasible adaptive policy remains the same as in this prior work under the conversion model?

**Limitations:**

This work does not have any societal impact. Limitations are well discussed.

**Strengths And Weaknesses:**

Strengths:

This paper is well written. It proposes a new structural model called conversion for contextual bandits with knapsacks inspired by work on logistic bandits. The model has interesting real-world applications. The regret metric and the proposed algorithm are intuitive. I think the main paper does a good job positioning the current work within the literature on contextual bandits with knapsacks. Comparison with existing works is sufficient.


Weaknesses:

The paper can do a better job of highlighting the technical novelty in the analysis. In the main paper, attention is paid to describing steps that bear similarities with the earlier works. Please elaborate if any non-trivial new technical machinery was introduced in the regret analysis apart from the KKT analysis.

A discussion on lower bounds and tightness of the results will help readers understand the merits of the proposed work better.

---

> ### Author Response · Authors · 2022-07-29
> **Answer to the weaknesses underlined and to the questions asked**
>
> We thank the reviewer for the careful reading and agree with the overall evaluation written. Interestingly, this overall evaluation (in terms of strengths and weaknesses) and the questions raised are extremely in line with what the other reviewers write!
>
> We would like to answer to the weaknesses underlined and to the questions asked.
>
> Weakness 1 (The paper can do a better job of highlighting the technical novelty in the analysis. In the main paper, attention is paid to describing steps that bear similarities with the earlier works. Please elaborate if any non-trivial new technical machinery was introduced in the regret analysis apart from the KKT analysis): Yes, we will do. The reviewer is right in spotting the KKT part (and the associated direct primal formulation of a strategy) as the main technical novelty in the analysis. To understand that this is the crux of our technical contributions, one must read Section E.3 in appendix, and admittedly, this should have been given more visibility, which we will do. In Appendix E.3, we relate our new approach to the approach by Agrawal and Devanur (2016). They had to rely on a parameter Z (to be learned), while we replace Z by dual optimal variables. Also (though we did not even mention it in the main body of the article) we think that we provide a cleaner and more transparent treatment of the use of the no-op action, which was often not specifically discussed (if not neglected) in earlier CBwK works. This weakness of not giving enough visibility to the new contributions was raised by the three reviewers, and we will edit the main body to reflect the true nature of our technical contributions, and in particular, move material from Section E.3 into the main body.
>
> Weakness 2 (A discussion on lower bounds and tightness of the results will help readers understand the merits of the proposed work better): We agree. For now, we mostly note (see comments after Theorem 2) that the orders of magnitude of our upper bound match the ones of previous bounds (obtained in different settings). The references cited actually only hint at some vague arguments tending to indicate that these orders of magnitude should be sharp enough, but none of them provides a clear and well-stated theorem with a lower bound. We actually had put the fact of stating a clean lower bound (together with other research perspectives) on our to-do list for a subsequent article. The present article, while being 47-page-long with all appendices, is but a milestone on the long and difficult road of the CBwK theory...
>
> Question 1 (Why negative regret appears in simulations?): Yes, we will comment on this. The expectation of the regret is non-negative, so the average regret computed on independent runs must be non-negative. However, due to to computational complexity and time-management issues, we were only able to compute 10 runs. We provided the confidence intervals on the associated expectation of the regret, and all contain 0 (and larger values), even when the empirical average over 10 runs is negative. We agree that this fact should have been commented and underlined. We will do so.
>
> Question 2 (When justifying comparing with the optimal static policy in regret, the work of Agrawal and Devanur, 2016, is mentioned. However, the setup here is not the same due to the conversion model. How is it ensured that the relation between the optimal static policy and the best feasible adaptive policy remains the same as in this prior work under the conversion model?): We are unsure to which part of the article this comment refers. We therefore provide two answers.
>
> Question 2--Answer 1, based on the text between line 126-129 in Section 2.1: The text is misleading indeed. We actually meant and should have written that the argument of Agrawal and Devanur (2016, Appendix B) can be adapted in a straightforward manner to prove this fact. Their argument is short (10-line-long) and easy to adapt. But it's true that given we provided lengthy and self-complete appendices, we could have well added a complete proof of this fact! We will do so.
>
> Question 2--Answer 2, why consider the policy by Agrawal and Devanur (2016) as a competitor in the simulations of Appendix F: In lines 1124-1129 in Appendix F, we agree that this policy is disadvantaged. This is even more true as it models rewards and costs independently, while they are coupled (and often simultaneously null). We nonetheless used this linear-modeling policy because a typical justification for linear approximations is that they are simple and work usually well. Also, we wanted to have some competitor to our policy (to which other policy could we compare it, given that we introduce a new setting?), and this one was only one easy to implement in practice---unlike the policies by Badanidiyuru et al. (2014) and subsequent works, which rely on considering finitely many benchmark policies. We will detail these issues more in the revision.

---

> > ### Comment · Reviewer_xxh2 · 2022-08-08
> > **Thank you for the response**
> >
> > My main concerns are addressed in the rebuttal. Please include the discussion on technical novelty, open questions regarding the lower bounds, experimental details regarding the benchmarks and new regret plots in the revised version.

---

### Official Review · Reviewer_fCTz · 2022-07-11

**Rating:** 6
**Confidence:** 2
**Soundness:** 3 good
**Presentation:** 2 fair
**Contribution:** 2 fair

**Summary:**

In this paper, the authors studied the contextual bandits with knapsack problem, where the rewards and costs are coupled through a conversion term. The authors proposed an adaptive policy for the proposed setting, and showed that it achieves a regret bound of order $\tilde{O}(\frac{\text{OPT}}{B} \sqrt{T})$, regardless of whether the context distribution is known or unknown. Finally, the authors also briefly discussed how their approach also applies to the linear contextual bandits with knapsack setting studied in [Agrawal and Devanur 2016].


**Questions:**

1. The authors mentioned that the regret bound in Theorem 2 can be further improved if the uniform deviation can be improved. Could you provide more details related to this statement? Why is the bound in equation (7) likely to be improved?
2. The authors state that the projection step in Phase 1 could be an issue. I wonder how this issue is dealt with? Do you do something different in numerical studies? Can you make certain changes to this step to improve the theoretical complexity?


**Limitations:**

None.

**Strengths And Weaknesses:**

Strengths:
1. The authors considered a novel setting that couples reward and cost using a conversion term, which corresponds to whether the customer accepts the offer. This is a realistic setting (though it will be ideal if the authors can move some of the discussion on real-world examples to the main body of the paper).
2. The authors proposed an adaptive policy for their setting and provided detailed discussion about each stage. They also provided theoretical results for the settings of both known and unknown context distributions and the proof appears sound.

Weaknesses:
1. The finiteness of the context set does appear limited. The authors mentioned that this is required only for Phase 2 of the algorithm and I wonder what are the theoretical results when one discretizes a continuous context set $\mathcal{X}$ for Phase 2?
2. Some numerical studies should be included in the main body of the paper. In particular, an illustration of the regret bound can be helpful in illustrating the dependency of the algorithm on $T$.
3. Including the lower bound results in the paper could be helpful, even though the authors referred to other works. Currently it is unclear how good the regret upper bounds are.

---

> ### Author Response · Authors · 2022-07-29
> **Answer to the weaknesses underlined and to the questions asked**
>
> We thank the reviewer for the careful reading and agree with the overall evaluation written. Interestingly, this overall evaluation (in terms of strengths and weaknesses) and the questions raised are extremely in line with what the other reviewers write!
>
> We would like to answer to the weaknesses underlined and to the questions asked.
>
> Weakness 1, first part (The finiteness of the context set does appear limited): Yes. But as we underline in Section 2.2, all earlier works on contextual bandits with knapsacks [CBwK] relied on heavy assumptions: either there was no constraint on the context set but regret was with respect to a finite number of pre-specified benchmark policies; or the cost and reward functions r and c were linearly linked to the contexts, i.e., the problem was parametric. In our case, while the conversion function P is parametric (but we do not require finiteness to learn its parameter \theta_\star in Pahse 1), we need some restriction or structural assumption on the functions r and c, and we picked finite contexts as such a restriction (so as to be able to carry out our Phase 2). The simulations of Appendix F (see in particular the end of Section F.2) discuss these issues further. This could certainly be improved one day: CBwK is an (extremely) difficult and largely unexplored problem, and admittedly, our contribution is only a milestone in developing a richer theory.
>
> Weakness 1, second part ([...] required only for Phase 2 of the algorithm and [...] theoretical results when one discretizes a continuous context set X for Phase 2?): They remain as stated in the main body of the article, though we did not underline this enough. We will do so in the revision (e.g., in Appendix F.2). More precisely, the error for learning \theta_\star (as outcome of Step 1 of the proof, see the equations of line 229) is then 'carried over' in the subsequent steps of the proof of Theorem 1 in a transparent way, as it is encompassed in the \epsilon terms. Finiteness of X plays a role in these subsequent steps only to be able to write and solve the convex optimization problems. So, P, as mentioned above, may well depend on non-finite contexts, but we have r and c take finitely many values by imposing that we cluster contexts into finitely many groups. Somehow, in practice, this only means that r and c are 'rougher' and 'more random' than we would like.
>
> Weakness 2 (Some numerical studies should be included in the main body of the paper. [...]): We agree... except that we did not find a way to provide a numerical study in 1 page. Appendix F reporting numerical results is 6-page-long: 1 page for the graphs but 5 pages to describe the setting (what contexts are, the functions P, c and r, etc.). Any hint/suggestion on this front would be much appreciated!
>
> Weakness 3 (Including the lower bound results in the paper could be helpful, even though the authors referred to other works. Currently it is unclear how good the regret upper bounds are): We agree with the fact that specific lower bounds should be proved. We mostly note (see comments after Theorem 2) that the orders of magnitude of our upper bound match the ones of previous bounds (obtained in different settings). The references cited actually only hint at some vague arguments tending to indicate that these orders of magnitude should be sharp enough, but none of them provides a clear and well-stated theorem with a lower bound. We actually had put this issue (together with other research perspectives) on our to-do list for a subsequent article. The present article, while being 47-page-long with all appendices, is but a milestone on the long and difficult road of the CBwK theory.
>
> Question 1: Yes, we suspect that the bound (7) could be slightly improved, but not in terms of the orders of magnitude in t (which is the critical point in the bound). It's merely that we apply T|X| times a simple Hoeffding-Azuma argument and that some more powerful empirical-process technique might lead to a slightly optimized bound, e.g., without a \sqrt{\ln T} term. But all this would only reduce slightly the b_T term in the bound of Theorem 2 and would not change the overall order of magnitude of the bound. We will ask some experts of empirical processes. We underlined this possibility of improvement only by honesty and to be transparent that we had not time to ask an expert, but it is really a detail (at least in our eyes).
>
> Question 2: Yes, and all earlier references on logistic bandits (Filipi et al. 2010, Faury et al. 2020) suffer from the same (deep) issue. However, we mention at the end of Section 3 that these references and our own simulations show that in practice, this projection step is virtually never required. We therefore did not bump into any problem when carrying out our numerical study, as the outcome of MLE was already in the required set \Theta. Admittedly, we took an \Theta large enough in our simulations.

---

> > ### Comment · Reviewer_fCTz · 2022-08-09
> > **thank you for your response**
> >
> > I thank the authors for their detailed response and they have addressed most of my questions on the technical results. In terms of future directions, I think the main limitations of the current work (e.g., finiteness of contextual space, lack of specific lower bound results, etc.) remain to be further looked at. A more clearly stated lower bound result, in particular, can be helpful for evaluating the performance of the proposed algorithm. Nevertheless, I think the current work proposed an interesting model along with solid technical results. Given its contributions and limitations, I have updated my score to a 6.

---

### Official Review · Reviewer_7uoE · 2022-07-14

**Rating:** 6
**Confidence:** 4
**Soundness:** 3 good
**Presentation:** 4 excellent
**Contribution:** 3 good

**Summary:**

This paper studied the problem of contextual bandits with knapsack constraints, where, in each round, given the stochastic i.i.d. context and the selected arm, a conversion rate is obtained. The reward and costs are coupled through the binary variable measuring conversion. The authors proposed a new adaptive policy that solves a linear program based on upper-confidence estimates of the probabilities of conversion in each round, and they show that the proposed policy achieves $(\mathrm{OPT}/B)\sqrt{T}$, where $B$ is the total budget allowed, $\mathrm{OPT}$ is the optimal expected reward, and $T$ is the number of rounds. The authors also showed that their proposed techniques can also be applied to the linear structures considered in [Agrawal and Devanur 2016].

**Questions:**

- For the projection step in Phase 1 of the proposed adaptive policy, could the authors elaborate on how the projection problem should be formulated and solved?

- The authors stated in quite a few places that their regret bounds can likely be improved. Could they provide a tighter analysis?

- The proposed adaptive policy and its analysis are based on or inspired by integrating several existing techniques (e.g., Logistic-UCB1 by [Faury et al. 2020], [Agrawal and Devanur 2016]). The key analysis steps are standard and similar to these existing works. Could the authors highlight the key differences, challenges, novelty, and contributions in their proofs?

**Limitations:**

This paper is theoretical and irrelevant to negative societal impact.

**Strengths And Weaknesses:**

Strengths:

+ This paper studied a new MAB model that integrates contextual stochastic bandits and knapsack constraints, which is well motivated in practice.
+ The authors proposed an adaptive policy that achieves $O(\sqrt{T})$ regret for the proposed contextual MAB model with knapsack constraints.
+ The proposed adaptive policy can also be applied to traditional linear contextual bandits with knapsacks.

Weaknesses:

- The proposed adaptive policy requires a finite context set.
- The computational complexity of the proposed adaptive policy is high: Phase 1 of the proposed policy requires a non-trivial projection step and Phase 2 assumes a finite context set.
- Theoretical analysis techniques are standard and some theoretical regret bounds are not sharp.

---

> ### Author Response · Authors · 2022-07-29
> **Answer to the weaknesses underlined and to the questions asked**
>
> We thank the reviewer for the careful reading and agree with the overall evaluation written. Interestingly, this overall evaluation (in terms of strengths and weaknesses) and the questions raised are extremely in line with what the other reviewers write!
>
> We would like to answer to the weaknesses underlined and to the questions asked.
>
> Weakness 1 (The proposed adaptive policy requires a finite context set): Yes. But as we underline in Section 2.2, all earlier works on contextual bandits with knapsacks [CBwK] relied on heavy assumptions: either there was no constraint on the context set but regret was with respect to a finite number of pre-specified benchmark policies; or the cost and reward functions r and c were linearly linked to the contexts, i.e., the problem was parametric. In our case, while the conversion function P is parametric (but we do not require finiteness to learn its parameter \theta_\star in Pahse 1), we need some restriction or structural assumption on the functions r and c, and we picked finite contexts as such a restriction (so as to be able to carry out our Phase 2). The simulations of Appendix F (see in particular the end of Section F.2) discuss these issues further. This could certainly be improved one day: CBwK is an (extremely) difficult and largely unexplored problem, and admittedly, our contribution is only a milestone in developing a richer theory.
>
> Weakness 2 (The computational complexity of the proposed adaptive policy is high [...]): For Phase 2, see the answer above. For Phase 1: yes, and all earlier references on logistic bandits (Filipi et al. 2010, Faury et al. 2020) suffer from the same (deep) issue. However, we mention at the end of Section 3 that these references and our own simulations show that in practice, this projection step is virtually never required.
>
> Weakness 3, first part (Theoretical analysis techniques are standard): Yes, all techniques taken in isolation are standard. The true technical contributions, which we admittedly did not underline enough in the main body of the article, would be that the algorithm directly uses a primal formulation. This was achieved thanks to the second step of the proof of Theorem 2, which looks so natural but follows from careful writing of all KKT conditions. That this is the crux of our technical contributions may be best understood by reading Section E.3 in appendix, where we relate our new approach to the approach by Agrawal and Devanur (2016). They had to rely on a parameter Z (to be learned), while we replace Z by dual optimal variables. Also (though we did not even mention it in the main body of the article) we think that we provide a cleaner and more transparent treatment of the use of the no-op action, which was often not specifically discussed (if not neglected) in earlier CBwK works. All in all, given that this weakness was raised by the three reviewers, we will edit the main body to reflect the true nature of our technical contributions, and in particular, move material from Section E.3 into the main body.
>
> Weakness 3, second part (Some theoretical regret bounds are not sharp): May we respectfully ask for more details on this? Yes, we suspect that the bound (7) could be slightly improved, but not in terms of the orders of magnitude in t (which is the critical point in the bound). It's merely that we apply T|X| times a simple Hoeffding-Azuma argument and that some more powerful empirical-process technique might lead to a slightly optimized bound, e.g., without a \sqrt{\ln T} term. But all this would only reduce slightly the b_T term in the bound of Theorem 2 and would not change the overall order of magnitude of the bound. We will ask some experts of empirical processes. We underlined this possibility of improvement only by honesty and to be transparent that we had not time to ask an expert, but it is really a detail (at least in our eyes).
>
> Question 1 (For the projection step in Phase 1 of the proposed adaptive policy, could the authors elaborate [...]?): We provide the formulation of the projection in Equation (4). Fortunately, it did not need to be solved in our simulations, as the outcome of MLE was already in the required set \Theta; see the answer to Weakness 2. Admittedly, we took an \Theta large enough in our simulations.
>
> Question 2 (The authors stated in quite a few places that their regret bounds can likely be improved): To the best of our knowledge, we only mention this for bound (7)---see the beginning and end of Section 5. See our answer above to Weakness 3, second part.
>
> Question 3 (The proposed adaptive policy and its analysis are based on or inspired by integrating several existing techniques. [...] Could the authors highlight the key differences [...]?): Yes, see the answer above to Weakness 3, first part. The main difference and contribution is w.r.t. Agrawal and Devanur (2016), who somehow used a dual formulation to be able to rely on online convex optimization techniques.

---

### Meta-Review · Area_Chair_mRq7 · 2022-08-24

**Recommendation:** Accept
**Confidence:** Certain

**Metareview:**

As acknowledged in the reviews (and I concur), this is a well written paper that introduces a relevant and interesting contextual-bandits model and gives solid technical results.  The paper certainly has its limitations, but overall the technical contributions are novel and well executed.  The authors have effectively addressed the questions and concerns brought up in the reviews.  I recommend acceptance.

**Award:**

No

---

### Decision · Program_Chairs · 2022-09-14

Accept